# Coaching doctors to improve ethical decision-making in adult hospitalised patients potentially receiving excessive treatment: Study protocol for a stepped wedge cluster randomised controlled trial

**Dominique D. Benoit**[1,2]\*, **Stijn Vanheule**[3], **Frank Manesse**[4,5], **Frederik Anseel**[3], **Geert De Soete**[3], **Katrijn Goethals**[6], **An Lievrouw**[2,7], **Stijn Vansteelandt**[8,9], **Erik De Haan**[10,11], **Ruth Piers**[1,12], on behalf of the CODE study group[13]¶

1 Ghent University Faculty of Medicine and Health Sciences, Gent, Belgium, 2 Intensive Care Medicine, University Hospital Ghent, Gent, Belgium, 3 Ghent University Faculty of Psychology and Educational Sciences, Gent, Belgium, 4 Independent, Conversio, Gent, Belgium, 5 Kets de Vries Institute, London, United Kingdom, 6 University Hospital Ghent Human Resources, Gent, Belgium, 7 Ghent University Hospital Cancer Centre, Gent, Belgium, 8 Faculty of Applied Mathematics, Computer Sciences and Statistics, Ghent University Faculty of Sciences, Gent, Belgium, 9 London School of Hygiene and Tropical Medicine, London, United Kingdom, 10 Hult International Business School Ashridge Centre for Coaching, Berkhamsted, United Kingdom, 11 VU Amsterdam School of Business and Economics, Amsterdam, The Netherlands, 12 Ghent University Hospital Geriatrics, Gent, Belgium, 13 University Hospital Ghent CODE Study Group, Gent, Belgium

¶ The complete list of the author group can be found in the Acknowledgments.
\* dominique.benoit@uzgent.be

## Abstract

### Background

Fast medical progress poses a significant challenge to doctors, who are asked to find the right balance between life-prolonging and palliative care. Literature indicates room for enhancing openness to discuss ethical sensitive issues within and between teams, and improving decision-making for benefit of the patient at end-of-life.

### Methods

Stepped wedge cluster randomized trial design, run across 10 different departments of the Ghent University Hospital between January 2022 and January 2023. Dutch speaking adult patients and one of their relatives will be included for data collection. All 10 departments were randomly assigned to start a 4-month coaching period. Junior and senior doctors will be coached through observation and debrief by a first coach of the interdisciplinary meetings and individual coaching by the second coach to enhance self-reflection and empowering leadership and managing group dynamics with regard to ethical decision-making. Nurses, junior doctors and senior doctors anonymously report perceptions of excessive treatment via the electronic patient file. Once a patient is identified by two or more different clinicians, an email is sent to the second coach and the doctor in charge of the patient. All nurses, junior and senior doctors will be invited to fill out the ethical decision making climate

**Data Availability Statement:** All relevant pilot data are within the manuscript and its Supporting

Information files. Deidentified research data will be made publicly available when the study is completed and published.

**Funding:** This study is supported by grants from the Ghent University Hospital "Fonds voor Innovatie en Wetenschappelijk Onderzoek" and the Belgian "Fonds voor Wetenschappelijk Onderzoek" FWO (senior clinical investigators grant 1800518N obtained by DB in 2017). The funders had and will not have a role in study design, data collection and analysis, decision to publish, or preparation of the manuscript.

**Competing interests:** The authors have declared that no competing interests exist.

questionnaire at the start and end of the 12-months study period. Primary endpoints are (1) incidence of written do-not-intubate and resuscitate orders in patients potentially receiving excessive treatment and (2) quality of ethical decision-making climate. Secondary endpoints are patient and family well-being and reports on quality of care and communication; and clinician well-being. Tertiairy endpoints are quantitative and qualitative data of doctor leadership quality.

## Discussion

This is the first randomized control trial exploring the effects of coaching doctors in self-reflection and empowering leadership, and in the management of team dynamics, with regard to ethical decision-making about patients potentially receiving excessive treatment.

## Introduction

### Background

The main goal of medicine is to reduce morbidity and mortality and to restore health without prolonging the suffering of patients. However, over the last decades the fast technical and medical progress has posed a significant challenge to doctors, having to find the right balance between life-prolonging and palliative care [1–4]. Several large multicenter studies suggest that clinicians in Belgium reflect less about the quality of care provided to patients with a high probability of dying (prospective study) and more often provide aggressive treatment at end-of-life (retrospective study) in comparison with clinicians in surrounding countries. In a study including patients who were 79±7 years old and died with cancer, Bekelman et al. found that 51% died in an acute care hospital and 17% were admitted in the ICU in the last 180 days of life in Belgium as compared to respectively 38% and 8% in Germany and 29% and 10% in the Netherlands [5]. De Roo et al. found that the percentage of time spent in the hospital in the last month of life in cancer patients was also the highest in Belgium (25%), followed by Italy (22%), Spain (18%) and the Netherlands (14%) [6]. These figures suggest that the trend in aggressive treatment at end-of-life together with the resulting shift in place of death (from home to care homes, to acute hospitals and finally to the ICU) that has been observed in the last decades in Western countries [5, 7–9] is more pronounced in Belgium than in the surrounding countries. This evolution contrasts with the preference of the majority of patients to die at home [9] in dignity and in the presence of their loved ones [9–11]. Belgium has one of the highest numbers of ICU beds per capita in Europe [12], while remarkably showing less openness to discuss ethically sensitive issues with patient and relatives, and among teams. For instance, Meeussen et al. found that end-of-life issues were less often discussed in the last month of life in cancer patients in Belgium (68%) than in the Netherlands (88%) [13]. End-of-life treatment preferences were also less often known (43% vs. 67%, respectively). In a recent prospective study performed in 68 ICUs across 12 European countries and the United States, our research group identified 4 mutually exclusive ethical decision-making climates via factor and cluster analyses of a 32-items survey; "good", "average with involvement of nurses at end-of-life", "average without involvement of nurses at end-of-life" and "poor" [4, 14]. In units with a poor ethical climate, there was less willingness to reflect on the quality of care through formal and informal team discussions. Dying was more often considered as therapeutic failure, and decisions in general and at end-of-life were more frequently postponed [14]. Decision-inertia was also objectively confirmed at the patient level in units with a poor climate, hereby validating our

ethical decision-making climate questionnaire (EDMCQ) instrument [4, 14]. Eight of the 13 participating ICUs in Belgium were classified as having poor climates (https://www.standaard. be/cnt/dmf20180606_03548418).

From the figures above, one may expect that clinicians are often confronted with patients potentially receiving excessive care, more specifically in Belgium. Three large multicentre studies have measured the prevalence of perception of excessive care in clinicians. In the APPRO-PRICUS study performed in 9 European countries and Israel, 27% of the ICU clinicians working on the day of study claimed to take care of at least one patient who received inappropriate, mostly excessive care [15]. Anstey et al. found a prevalence of 38% among 1363 clinicians working in 56 Californian ICUs. In contrast to the APPROPRICUS study, these investigators found a difference in prevalence between doctors and nurses (51% vs. 36%, respectively p<0.001) [16]. Benoit et al. recently assessed the incidence of perceptions of excessive care by clinicians in patients admitted in 68 ICUs in Europe and the United States [14]. Of the 1761 patients admitted over a 28-day period, 369 (20.9%) and 181 (10.2%) were identified as receiving excessive care by at least one and by at least two clinicians, respectively. Whereas patients perceived as receiving excessive care by two or more clinicians had a 7% probability of surviving at home with a good quality of life at one year, the probability of receiving a written do-not-intubate and resuscitate (DNIR) order was 30% only, varying from 35% in units with a good versus 20% in units with poor ethical decision-making climate (p = 0.011). In order to perform the power analysis for the current study, we measured the incidence of patients with two or more perceptions of excessive care by different clinicians in the wards of the Ghent University Hospital willing to participate in this study. A dedicated researcher actively surveyed all nurses and doctors during one week on every department asking for which patients the clinician was responsible and in which patients they perceived the care as excessive. The DNIR code for these patients was retrieved from the head nurse. Of the 258 patients admitted in these wards, 32 (12%) were perceived as receiving excessive care by two or more different clinicians and only 12 (38%) had a written DNIR order (Table 1). This is in line with the ICU setting [14]. The incidence of patients perceived as receiving excessive care by two or more

**Table 1. Incidence of patients with two or more perceptions of excessive treatment and do-not-intubate and resuscitate order in the 10 participating departments over one week pilot study (conducted in July 2019).**

| Department | n (%) with 2 PETs | n (%) of 2 PETs with DNIR code |
|---|---|---|
| 1 | 3/32 (9%) | 2/3 (67%) |
| 2 | 4/19 (21%) | 2/4 (50%) |
| 3 | 5/26 (19%) | 1/5 (20%) |
| 4 | 2/40 (5%) | 1/2 (50%) |
| 5 | 8/34 (24%) | 2/8 (25%) |
| 6 | 2/18 (11%) | 1/2 (50%) |
| 7 | 5/58 (12%) | 1/5 (20%) |
| 8 | 3/41 (7%) | 2/3 (67%) |
| Total | 32/258 (12%) | 12/32 (38%) |
| 9* | | |
| MICU** | 10% | 34% |

Abbreviations: PET: perceptions of excessive treatment, DNIR: do-not-intubate and resuscitate, MICU: medical intensive care unit

In total, 3379 perceptions of excessive care were collected in 3703 clinicians. The response rate was 91%

* The incidence was not measured in department 9.

** The incidence in the medical intensive care unit was retrieved from the Dispropricus database [14].

clinicians and written DNIR order in these patients varied from 5% to 24% and from 20% to 67% between departments, respectively. This means that the ethical principle of not harming was potentially violated in 20 patients admitted in these wards during that week given the risk of receiving cardio-pulmonary resuscitation or of being referred to the ICU in case of deterioration despite the poor expected outcomes at one year [17, 18].

In addition to potentially violating the basic bioethical principles, excessive treatment may increase the risk of physical and psychological burden in patients and relatives [2–4, 19–22]. Excessive treatment may also induce moral distress, compassion fatigue or avoidance behavior in clinicians [2–4, 23] with conflicts, or even worse, chronic animosity and distrust in the team as a consequence [2–4, 23–29], which will not benefit to the patient [4, 25, 26]. These issues may be even more pertinent considering the high number of patients potentially receiving excessive treatment against their or relatives' wishes [14], or receiving treatment that is potentially discordant with their written-treatment-limitation decision [30]. Furthermore, postponing end-of-life decision-making and a high mortality in the ward have been associated with burnout and a higher intent to leave in clinicians [2–4, 23, 31–34]. Finally, providing excessive treatment also has financial consequences for relatives [35] and for society [36–38].

In conclusion, literature and a pilot study suggest a need for openness in discussing ethically sensitive issues within and among teams, and for improving decision-making for the benefit of the patient at end-of-life, worldwide and more specifically in Belgium. The current intervention aims at achieving this objective at the Ghent University Hospital in Belgium by coaching doctors in self-reflective and empowering leadership, and in the management of team dynamics via repeated discussion of actual patient situations identified by clinicians in daily care. Care (on a meta-level) can mean more or less treatment, depending on the circumstances. One can never offer too much care in this sense. Hence we will use the term 'excessive treatment' throughout, in line with a recent expert consensus meeting [39].

## Rationale

To improve care at the end-of-life for seriously ill patients and their families, many research groups around the world focus on advance care planning, and earlier integration of palliative care. To our knowledge, our research group is the first to focus on the value of subjective impressions (perceptions) of clinicians in addition to objective criteria to identify timely patients potentially receiving excessive treatment.

We chose this approach more than a decade ago [15] because of the following reasons:

1. Although the medical community puts tremendous efforts in trying to improve prognostication via objective factors or scoring systems [40], their predictive value is becoming less important because medical and technical innovations frequently exclude patients' spontaneous death. Nowadays, patients die mainly after doctors have decided to withhold or withdraw treatment together with the patient or their relatives [2, 3, 39]. Moreover, prognostic criteria or scoring systems fail to predict outcome at the individual patient level [1, 40].

2. Despite the availability of universal objective prognostic factors for many diseases, a large variability in written DNIR orders, utilization in health care resources at end-of-life, palliative care and place of death have been observed across continents, countries, hospitals, wards, doctors and patients [5, 41–48], even after adjustment for the case-mix. This indicates that subjective factors at the personal ("style"), team ("climate") and country ("culture") level are more important than objective factors during ethical decision-making. However, subjective factors are rarely acknowledged and expressed by clinicians, more specifically by doctors at the bedside [4, 24, 25].

3. Focusing tenaciously on objective criteria only, bears witness to a defensive avoidance strategy that draws attention away from difficult and aversive care-related situations that might provoke anxiety [25–27], like end-of-life situations, which require ethical decision-making. Nowadays, this is potentially even more problematic because of point 1.

4. Focusing tenaciously on objective criteria puts accountability outside oneself and may therefore retain doctors from acting, whereas subjective criteria should be integrated in the (shared) decision-making process for the benefit of the patient [4, 14, 25–27].

5. Balanced medical ethical decision-making and empowering leadership should take into account both objective information and subjective criteria including goals, emotions and values of patients and relatives [4, 14, 25, 26], and assessments of all parties in the multidisciplinary team [3, 4]. Although the four ethical principles (beneficence, non-maleficence, autonomy and distributive justice) and their critical considerations remain essential, meaning-making through human stories (narrative ethics) and shared understanding of patient's situations through dialogue (hermeneutics) [4, 49, 50] is today all the more important because of point 1.

6. Difficult aversive care-related situations trigger professionals' subjective goals, emotions and values which may lead to disagreement [2–4, 24–27] and avoiding ethical decision-making, thus undermining empowering leadership. Therefore, group dynamics concerning decision-making and leadership in medical teams should be monitored and discussed explicitly.

7. Subjectivity is prone to bias. Nevertheless, it is of significant prognostic value, more specifically when expressed by several clinicians. As mentioned above, the probability of being alive, at home with a good quality of life one year after ICU admission was only 7% in patients who were perceived as receiving excessive treatment by two or more clinicians in the multicentre DISPROPRICUS study [14].

*Because of all these reasons, concordant perceptions of excessive treatment by two or more clinicians can be considered as an ideal palliative care trigger, a signal that the team should reflect on the quality of care that is provided to the patient and whether the treatment is in balance with the medical condition of the patient and the patient's goals of care.* Moreover, this approach has the advantage of identifying excessive treatment on "human grounds" rather than on abstract objective criteria. This further stimulates engagement and accountability in clinicians, more specifically in nurses and junior doctors. However, creating a climate that enables clinicians to speak up without fear for a verbal or non-verbal reprimand, or being considered as "incompetent" or "not respectful" towards higher ranked professionals like doctors, seems a precondition [4]. The key position of doctors in the ethical decision-making process and the fact that senior doctors tend to overrate their communication, leadership- and decision-making capacities in general [51] and more specifically at end-of-life [14, 52–54] naturally points to them for these interventions. This pattern might be even more pronounced in units with a poor ethical climate [14, 53].

We aim at creating a safe climate that enhances inter-professional shared decision-making for the benefit of the patient and at enhancing specific self-reflective and empowering leadership skills (including the management of group dynamics in the interdisciplinary team) [4, 14, 55–58]. Probably, these skills will also help doctors during patient and family meetings, and enable them to better take into account the patient's and family's wishes.

For ease of read, patients who are perceived as receiving excessive treatment by two or more clinicians will from now on be denominated as patients potentially receiving excessive treatment given their poor prospects [14].

## Objectives and study hypotheses

The primary objective of this study is to investigate whether coaching doctors during 4 months in self-reflective and empowering leadership and in managing team dynamics with regard to adult hospitalized patients potentially receiving excessive treatment improves ethical decision-making in comparison with usual care. Except from a treatment-limitation-decisions guideline which focuses on the legal and deontological framework, no other guideline with regard to ethical decision-making has been implemented at the Ghent University Hospital. In one ward (geriatrics), there is a ongoing project in which a clinical nurse specialist stimulates and performes advance care planning conversations with patients and/or relatives at request of the team and who organizes debriefings when needed based on the ethical concerns of the nurses.

The quality of medical-ethical decision-making will be assessed *objectively* via the incidence of written DNIR orders in patients potentially receiving excessive treatment between hospital admission and the end of the first hospital stay (first primary endpoint) and *subjectively* via the EDMCQ [4] that will be filled out by the doctors and nurses in the team at the start and end of the 12 months study period (second primary endpoint) (Fig 1). The study is planned to start on the 10th January 2022.

These primary objectives can be formalized in the following two study hypotheses:

- The intervention changes the incidence of advance care planning operationalized by written DNIR order in adult hospitalized patients potentially receiving excessive treatment from 35% (under usual care) to 50% over the 12-month study period.

- The intervention increases the average EDMCQ score in clinicians (doctors and nurses) by 2.8 points (equals the sum of the differences in the 7 factors between units with "good" and with an "average ethical climate with involvement of nurses at end-of-life" in the DISPRO-PRICUS study [14]) over the 12-months study period.

By evaluating both primary endpoints we aim to assess whether our intervention has an effect on individual decision-making by doctors and/or on collective decision-making in team (Fig 2). Both null hypotheses that we test express no change (as opposed to change).

The secondary objective of this study is to investigate whether the intervention improves communication with patients and relatives, and reduces the burden of patients, relatives, clinicians and costs (see Tables 2 and 3) in comparison to usual care.

The tertiary objective of this study is to investigate the relationship between leadership styles and coachability of the doctors according to the coaches at the primary and secondary endpoints. For this purpose, quantitative and qualitative data of doctor leadership quality will be collected (see Table 4).

## Materials and methods

### Trial design

The study follows a stepped wedge cluster randomized trial design, run across 10 different departments of the Ghent University Hospital. All 10 departments were randomly assigned to start a 4-month coaching period in month $k = 1,\ldots,10$ following a stratified design. In particular, the 3 departments with the highest incidence of written DNIR orders (based on historical data, Table 1) were randomly assigned to start the intervention in months 2, 4 and 6 (each time together with another ward). The 7 other wards were randomly assigned to start the intervention according to the schematic overview in Fig 3. Subsequently, departments in which senior doctors remain in charge of their own hospitalized patients (in contrast to departments in which one senior doctor is in charge of all hospitalized patients on a specific ward) were

| Timepoint | T0 | T1 | T2 | T3 | T4 |
|---|---|---|---|---|---|
| EDMCQ by nurses, junior and senior doctors | | X | X | | |
| Department characteristics by head nurse and medical head of department | X | | | | |
| Daily perceptions of nurses, junior, senior doctors | | ————————————————→ | | | |
| Inclusion of patients identified as potentially receiving excessive care | | ————————————————→ | | | |
| Datacollection* in patients and relatives | | ————————————————→ | | | |
| Datacollection in medical charts | | ————————————————→ | | | |
| Evaluation of the intervention in participants (survey and focus groups) | | | | | X |

T0: start of the study period of 12 months
T1: start of the intervention period (depends on the randomization per team)
T2: end of the intervention period of 4 months (depends on the randomization per team)
T3: end of the study period of 12 months
T4: post-intervention (depends on the randomization per team)

*Patients and their families are asked
- during hospitalization: informed consent and basic characteristics during hospitalization
- 3 weeks after discharge: survey on wellbeing, rating of quality of care, communication and decision-making
*Patients are asked
- 1 year after discharge: survey on living situation
EDMCQ: ethical decision-making climate questionnaire

**Fig 1. Schedule of enrolment, interventions and assessments.**

spread in order to reduce the workload of the coach. One month was added to compensate for the absence of the coach for whatever reason. All wards will be followed in terms of the primary and secondary endpoints over the 12-month duration of the study.

## Recruitment and eligibility

All departments within Ghent University Hospital frequently referring patients to the ICU were invited to participate in the current study during team meetings organized in 2018–2019. All 10 participating departments acknowledge room to improve their ethical decision-making and were enthusiastic to participate in this study. These departments are: Cardiology, Gastroenterology and Hepatology, General Internal Medicine, Geriatrics, Hematology, Medical Oncology, Neurology, Nephrology, Pulmonology and the Medical ICU. Surgical departments did not express the need to improve ethical decision-making in our hospital. Because of financial constraints the department of pediatrics could not be included in the study. However, the pediatric neurology and pulmonology departments were included in October 2021 in a pilot study to test the intervention and procedures.

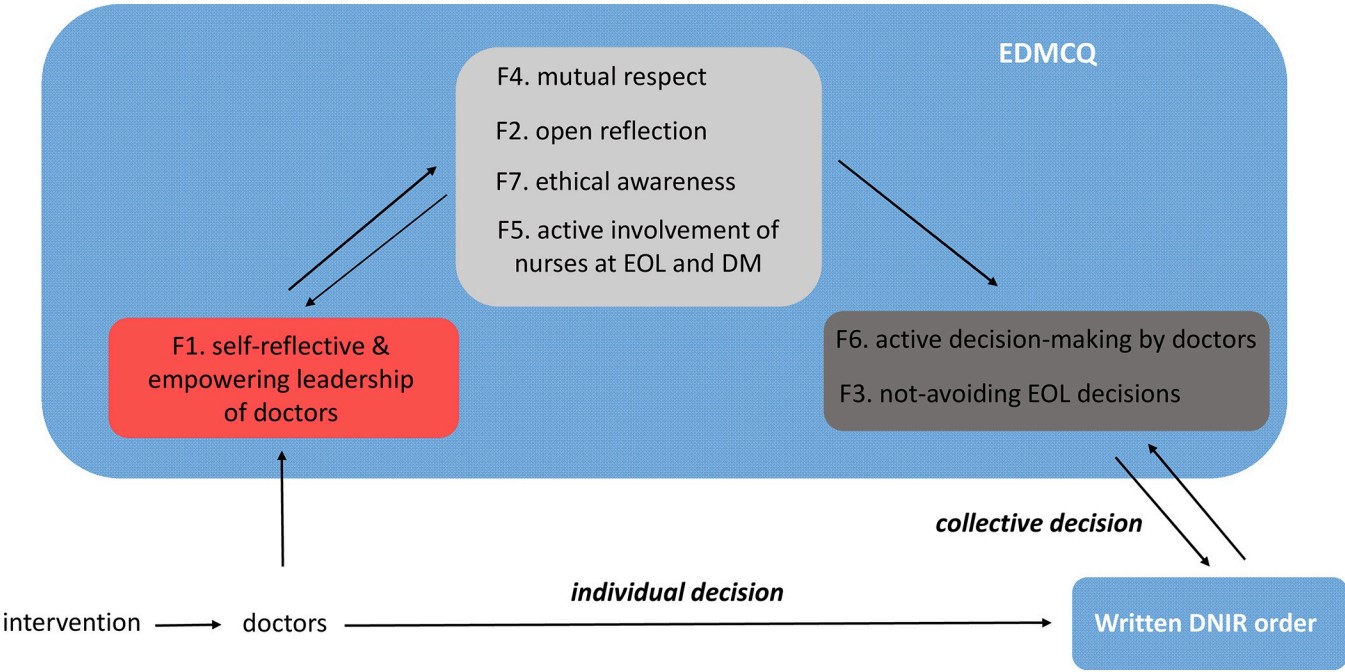

**Fig 2. Theoretical background on the primary endpoints.** The quality of medical ethical decision-making will be assessed objectively via the incidence of written Do-Not-Intubate and Resuscitate (DNIR) orders in patients potentially receiving excessive treatment during their first hospitalization and subjectively via the Ethical Decision-Making Climate Questionnaire (EDMCQ) [4] that will be filled out by the doctors and nurses in the team one month prior and after the 12-months study period. The EDMCQ is 32-item validated questionnaire consists of 7 main domains or factors: factor F1 "self-reflective and empowering leadership of doctors", F2 "open and interdisciplinary reflection", F3 "not avoiding end-of-life decisions", F4 "mutual respect within the interdisciplinary team", F5 "active involvement of nurses in end-of-life care and decision-making", F6 "active decision-making by doctors", F7 "ethical awareness". We expect that an effect on individual decision-making by doctors would affect the incidence of written DNIR orders both directly, and indirectly via F3 and F6. We expect that effect on collective decision-making in team would affect F1, which in turn affects all other EDMCQ factors and via these, the incidence of written DNIR orders. Both null hypotheses that will be tested express no change (as opposed to change).

All junior and senior doctors taking care of adult (> 18 year old) hospitalized patients in the 10 participating departments of the Ghent University Hospital are eligible for the intervention. Dutch speaking patients and one of their relatives will be included for data collection at the patient, relative and societal level. Besides identifying patients potentially receiving excessive treatment during the 12 months study period, all nurses, junior and senior doctors will be invited to fill out the EDMCQ.

## Intervention

The CODE intervention in junior and senior doctors consists of 4 components:

1. One interactive session of 2–3 hours focusing on the concepts of medical-ethical decision-making, the psychological challenge of dealing with ethically sensitive medical topics, and empowering leadership.

2. Observation of interdisciplinary team meetings (where patient care is discussed among doctors, nurses and other professionals) by a first coach who also gives feedback to the doctor in charge to enhance self-reflection on empowering leadership and managing group dynamics during the 4-month intervention period.

3. Individual coaching by the second coach, focusing on self-reflective and empowering leadership, as well as on managing group dynamics with regard to ethical decision-making

**Table 2. Overview of collected data and timelines: Primary outcomes and secondary outcomes on patient level.**

| OUTCOME DOMAIN | Outcomes | Instrument | Data Source | Timing |
|---|---|---|---|---|
| **Ethical decision-making (primary endpoints)** | | | | |
| ADVANCE CARE PLANNING | Incidence of written DNIR | | Chart extraction | Up to the end of the first hospital stay |
| TEAM PERCEPTION OF ETHICAL CLIMATE | Ethical decision-making climate | EDMCQ | Nurses and physicians | At the start and the end of the 12-months study period |
| **Patient-centered outcomes (secondary endpoints)** | | | | |
| LIVING SITUATION AT 1 YEAR | Incidence of death | survey | Patient/family survey by mail or phone | 1 year after first hospital admission |
| | Combined endpoint: dead, not at home or utility score < 0.5 | Euro-QOL+survey | Patient/family survey by mail or phone | 1 year after first hospital admission |
| LENGTH OF HOSPITAL STAY | Number of days admitted in the hospital | | Chart extraction | At the end of the first hospital stay |
| QUALITY OF CARE, COMMUNICATION AND DECISION-MAKING | Pain | NRS | Chart extraction | Up to the end of the first hospital stay |
| | Satisfaction | Euro-FS adapted for the patient | Patient discharged alive | 3 weeks after hospital discharge |
| | Potentially inappropriate or burdensome treatments | Number of potentially inappropriate or burdensome treatments* | Chart extraction | Up to the end of the first hospital stay |
| WELL-BEING AFTER DISCHARGE | Anxiety and depression | HADS | Patient discharged alive | 3 weeks after hospital discharge |
| QUALITY OF DYING IN PATIENTS WHO DIED ON THE WARD | Quality of dying | QODD-nurse | Nurse | Within 1 week after patient's death |
| | | Euro-QODD | Family | 3 weeks after patient's death |

Abbreviations: DNIR = Do-Not-Intubate and Resuscitate, EDMCQ = Ethical Decision-Making Climate Questionnaire, QOL = quality of life, NRS = Numeric rating scale, FS = Family satisfaction, EOL = End-of-life, ICU = Intensive care unit, HADS = Hospital anxiety and depression scale, QODD = Quality of dying and death questionnaire

* resuscitation, ICU admission, (non-) invasive ventilation, dialysis, surgical procedures, chemotherapeutic and radiotherapeutic treatments

EDMCQ is a validated questionnaire consisting of 7 main domains or factors: F1 "self-reflective and empowering leadership of doctors", F2 "open and interdisciplinary reflection", F3 "not avoiding end-of-life decisions", F4 "mutual respect within the interdisciplinary team", F5 "active involvement of nurses in end-of-life care and decision-making", F6 "active decision-making by doctors", F7 "ethical awareness". Factor scores on 7 domains, which is normally distributed, centred at mean of zero, with standard deviation 5.5 (minimum score -25, maximum score 25). Higher scores indicate higher quality of ethical decision-making.

Euro-QOL-5D measures health-related quality of life, with possibility of conversion of each health state in a utility index (range -0.1584 to 1.000). This questionnaire measures health in five dimensions: mobility, self-care, usual activities, pain/discomfort, and anxiety/depression [66].

NRS for assessment of pain ranging from 0 (no pain) to 10 (worst possible pain) [69]. First endmpoint concerning pain is sum of the average daily score up to the end of the first hospital stay, second endpoint is number of days with an average score more than 3.

Euro-FS is a validated 18-item questionnaire covering satisfaction with 4 domains: communication, empathy, symptom management and decision-making [67, 68]. We will use the single item assessment of satisfaction of this score ranging from 0 to 10. Higher values indicate higher satisfaction with care.

HADS is a 14-item self-report assessment with subscales for anxiety and depression. Each domain has a score range of 0–21 with the following interpretation: 0–7 normal, 8–10 mild, 11–21 moderate to severe [70].

QODD-nurse: the validated Dutch version of the QODD instrument is a validated 22-item questionnaire. Euro-QODD family is a 14 item questionnaire to allow families to assess patients' quality of dying and death [68]. For QODD-nurse and Euro-QODD family, we will use the single-item assessment of quality dying and death of this score ranging from 0 to 10. Higher values indicate higher quality.

about patients who are perceived to receive excessive treatment during the 4 months intervention period, and in absence of such patients, focusing on all aspects of ethical decision-making that are important for the coachee. Every doctor will be invited to participate in at least 8 coaching sessions of 1 hour during the intervention period, to be extended on request.

Coaching will be provided by senior clinical psychologists who are trained in coaching.

**Table 3. Overview of collected data and timelines: Secondary outcomes on family, clinician and societal level.**

| OUTCOME DOMAIN | Outcomes | Instrument | Data Source | Timing |
|---|---|---|---|---|
| **Family (secondary endpoints)** | | | | |
| WELL-BEING AFTER DISCHARGE | Anxiety and depression | HADS | Family | 3 weeks after first hospital discharge |
| | Post-traumatic stress | IES-R | Family of patients who died on the ward | 3 weeks after patient's death |
| QUALITY OF COMMUNICATION AND DECISION-MAKING | Satisfaction | Euro-FS | Family | 3 weeks after first hospital discharge |
| **Healthcare utilization (secondary endpoints)** | | | | |
| | Payer's hospitalization cost | | Hospital billing record | End of first hospitalisation |
| | Number of medical interventions* | | Chart extraction and patient/family by telephone call | 1 year after first hospital discharge |
| **Clinicians (secondary endpoints)** | | | | |
| WELL-BEING ON INDIVIDUAL LEVEL | Stress (mild-moderate-severe-extreme) related to perception of excessive treatment | | Nurses and physicians | At the start and end of the 12-months study period |
| | Intention to leave job | | Nurses and physicians | At the start and end of the 12-months study period |
| WELL-BEING ON TEAM LEVEL | Sick leave | | HR department | At the start and end of the 12-months study period |
| TEAM PERFORMANCE | Ethical practice score | Ethical practice score | Head nurse | At the start and end of the 12-months study period |

Abbreviations: HADS = Hospital anxiety and depression scale, IES-R = Impact of events scale revised, FS = Family satisfaction, ICU = intensive care unit, HR = Human resources

* emergency department visits, hospital admissions, ICU admissions, total days in hospital, total days in ICU, dialysis, surgical procedures, chemotherapeutic and radiotherapeutic treatments, blood analyses, radiologic investigations

HADS is a 14-item self-report assessment with subscales for anxiety and depression. Each domain has a score range of 0–21 with the following interpretation: 0–7 normal, 8–10 mild, 11–21 moderate to severe [70].

IES-R is a 22-item scale to self-report the frequency of intrusive and avoidant phenomena after a variety of traumatic experiences [72]. Total stress score is interpreted as follows: low risk for PTSD (0–11), moderate risk (12–32), high risk (33 or higher).

Euro-FS is a validated 18-item questionnaire covering satisfaction with 4 domains: communication, empathy, symptom management and decision-making [67, 68]. We will use the single item assessment of satisfaction of this score ranging from 0 to 10. Higher values indicate higher satisfaction with care.

Ethical practice score consists of 12 items. We will use the 10 department specific items (minus the 2 country-specific items). The score ranges from 0 to 10 with higher scores indicating a higher degree of ethical practice organization.

Both coaches receive supervision provided by an internationally recognized coaching expert and trainer.

We refer to S2 Appendix for a more extended coaching protocol consisting of a) the focus of the study, b) the detailed methodology used by the coaches during the intervention, c) the quality assurance process used during supervision and d) the references to the international standards.

4. During the intervention coaches and doctors in charge will be informed of the presence of a patient potentially receiving excessive treatment in their ward, by an electronic alert. All clinicians will be invited to provide a perception of excessive treatment via the EPD when they feel that the treatment provided to their patient is excessive. Excessive treatment is defined as treatment that is perceived to be no longer consistent with the expected survival or quality of life ("too much treatment") or that is provided against the patient's or relatives' wishes [14, 15]. Anonymity of the clinicians will be guaranteed.

**Table 4. Overview of collected data and timelines: Tertiary outcomes\*.**

| OUTCOME DOMAIN | Outcomes | Instrument | Data Source | Timing |
|---|---|---|---|---|
| **REACH** | Inclusion rate and number of coaching sessions | Diary | Coach | During the 4 months intervention period |
| **EFFICACY** | Self-reflective and empowering leadership skills | Survey on quality of self-reflection and empowering leadership | Coach | During the 4 months intervention period after every coaching session |
|  | Competencies in empowering leadership and management of team dynamics | Survey on quality of empowering leadership and management of team dynamics | Observator of interdisiplinary team meeting | During intervention after every team meeting observation |
| ADOPTION, IMPLEMENTATION AND MAINTENANCE\*\* | Experiences and satisfaction with the intervention; fidelity to the protocol and long-term perspectives | Focus groups | Doctors and nurses | 3 months after the intervention |
|  |  | Survey (user experience) | Doctors | 3 moths after the intervention |

\*Process measures based on RE-AIM framework

\*\* Adoption = experiences with the intervention, Implementation = feasibility, acceptability (experiences and satisfaction) and fidelity to the protocol,

Maintenance = long-term adaptation of the intervention

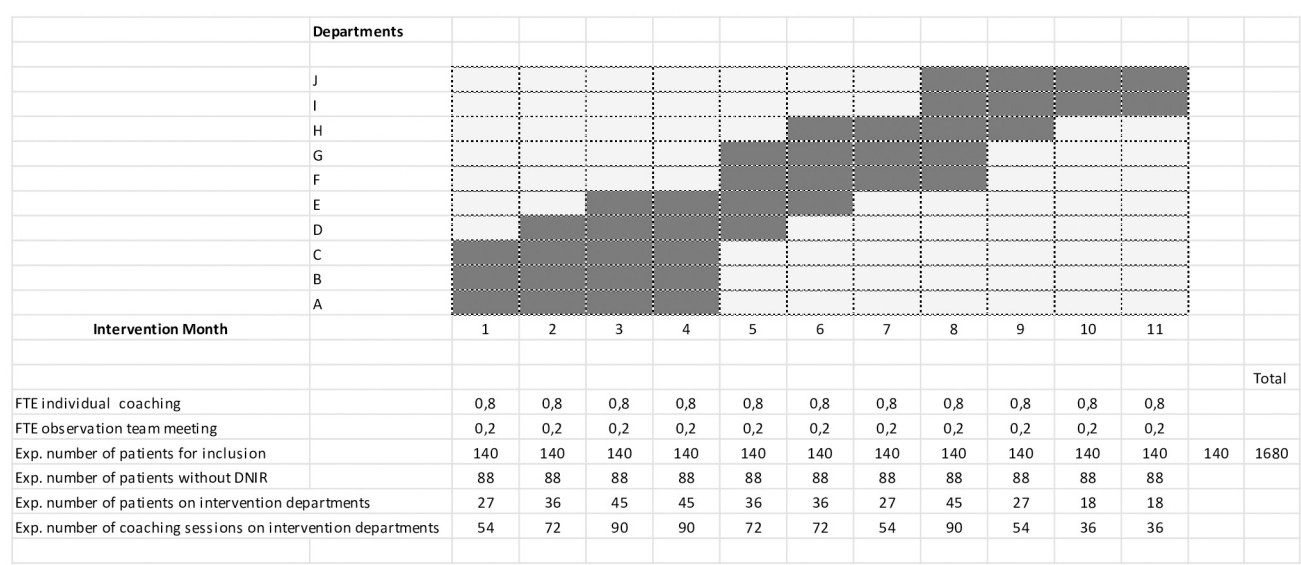

**Fig 3. Trial design (stepped wedge cluster randomized controlled trial).** Based on the results of a pilot study (Table 1), we expect that 4 patients per ward per week will be identified by clinicians as potentially receiving excessive care; expanded to 1 month and 10 wards this yields to 160 patients. Taking to account that one in eight patients might be re-admissions and/or patients with length of stay of more than one week, we drop this total amount to 140, of whom 88 without a DNIR code over 10 departments. This yields to 9 patients and 18 coaching sessions (one junior and one senior doctor per patient) per month per department. As 2 to 5 departments will have the intervention at the same time, the expected number of coaching sessions varies between 36 and 90 sessions per month. The black bars indicate the intervention period across all 10 departments. All 10 departments were randomly assigned to start a 4-month coaching period in month k = 1,...,10 following a stratified design. In particular, the 3 departments with the highest incidence of written DNIR orders (based on historical data, Table 1) were randomly assigned to start the intervention in months 2, 4 and 6 (each time together with another ward). The 7 other wards were randomly assigned to start the intervention. Subsequently, departments in which senior doctors remain in charge of their own hospitalized patients (in contrast to departments in which one senior doctor is in charge of all hospitalized patients on a specific ward) were spread in order to reduce the workload of the coach. One month was added to compensate for the absence of the coach for whatever reason.

## Outcomes and timeline

Besides the incidence of written DNIR-orders and the EDMCQ [4], the following outcomes will be collected (Tables 2–4):

**Patients' level.** Incidence of death one year after the first hospital admission will be collected as one of the secondary endpoints. However, because staying at home with a good quality of life is highly valued by patients, the combined one-year patient outcome in this study will be defined as deceased, not at home or a utility score < 0.5 [14]. For the combined endpoint, Euro-QOL-5D measures health-related quality of life, with possibility of conversion of each health state in a utility index (range -0.1584 to 1.000) [59]. Also, number of days of hospital admission will be assessed.

For quality of care and communication and decision-making, we will use the Euro-FS (European Family Satisfaction in the ICU) [60, 61].

We will use the NRS score for pain as objective measure for symptom control [62].

For well-being, we will use HADS (Hospital anxiety and depression scale) which is a validated 14-item self-report assessment with subscales for anxiety and depression [63].

For quality of dying in patients who died on the ward, we will use the QODD (Quality of dying and death questionnaire) [64].

For the number of potentially inappropriate and burdensome treatments, we will collect resuscitation, ICU admission, (non-) invasive ventilation, dialysis, surgery, chemotherapeutic and radiotherapeutic treatments.

**Relatives' level.** Same as in patients, we will use Euro-FS for quality of care, communication and decision-making and HADS for family well-being.

For psychological burden of families who lost their relative on the ward, we will use the IES-R (Impact of events scale-revised) [65].

**Clinicians' level.** For clinician well-being, we will use moral distress (stress related to the perception of excessive treatment), sick leave(team-level), intentional job-leave (individual level) as endpoints [34]. For team performance we will use the ethical practice score.

**Societal level.** For hospital costs, we will assess hospital costs by the hospital billing record. We will collect emergency room visits, hospital and ICU admissions and number of diagnostic procedures (dialysis, surgical procedures, chemotherapeutic and radiotherapeutic treatments, blood analyses and radiological investigations) in the 12 months after the first hospital discharge.

## Sample size

The written DNIR order analysis will be based on logistic mixed effect models with random intercept to account for between-department variability, assuming a constant risk before and after intervention, and a linearly changing risk during the intervention. Based on such analysis, under the previously described stratified randomized design, a Monte Carlo power evaluation showed that a Wald test at the 5% significance level delivers 86% power to detect an intervention effect when data are available for 5 patients potentially receiving excessive treatment per department per month (over a period of 12 months), if the risk of written DNIR order in patients potentially receiving excessive treatment increases from 35% before to 50% after intervention; with an incidence of patients potentially receiving treatment equaling 12% (Table 1), this amounts to approximately 42 patients per department per month or 5040 patients in total over 10 departments and 12 months, of which 605 (12%) potentially receive excessive treatment.

Based on our pilot measurement (Table 1), we expect that 16 patients per ward per month will be identified by clinicians as potentially receiving excessive treatment. Taking into account

that one in eight patients might be re-admissions and/or patients with length of stay of more than one week, we reduce this to 140 over the 10 wards, yielding an expected 1680 patients potentially receiving excessive treatment over a 12-month period which far exceeds the required sample size. Based on the data from the pilot study, we first fitted a logistic mixed effects model for the incidence of excessive treatment. This resulted in a between-unit standard deviation of 0.25, corresponding to an intraclass correlation of 0.025, used for the power analysis.

In a secondary analysis, we will additionally adjust for a fixed linear time slope to account for period effects, even though no such effects are expected. With 16 patients potentially receiving excessive treatment per department per month (over a period of 12 months), this secondary analysis has 70% power at the 5% significance level to detect an intervention effect under the aforementioned conditions.

For the EDMCQ, the sample size calculation was based on linear mixed models for the change in EDMCQ score after versus before the intervention, including a random intercept to account for between-department variability and a random intercept for variability between clinicians. Based on such analysis, a Monte Carlo power evaluation showed that a Wald test at the 5% significance level delivers 93% power to detect an intervention effect when data are available for 5 clinicians per department, if the EDMCQ score increases on average with 2.8 units. For this, we assumed an intra-department correlation of 0.14, an intra-clinician correlation of 0.25 and a total standard deviation of 5.03. Additionally adjusting for a fixed linear time slope to account for period effects reduces power to 39%.

The software used for power calculation was RStudio 2022.07.2+576 "Spotted Wakerobin" for macOS Mozilla/5.0.

## Randomisation

Randomisation of the 10 departments was performed by the Ghent University Department of Applied Mathematics, Computer Science and Statistics based on a random number generator in the software R. As in nearly all stepped wedge designs, the nature of the intervention is such that it cannot be blinded to clinicians. However, clinicians will be asked to blind the patients to the time of the intervention.

## Data collection

**Patient and family data.** All adult patients (or their legal representative in case of incompetence) identified as potentially receiving excessive treatment will be asked written informed consent by the treating physician to fill-out a survey 3 weeks after discharge and to be contacted 1 year after discharge. This survey can be taken by mail or by phone. They will be asked to indicate 1 family member to be contacted. That family member will be asked written informed consent also during hospitalization. The investigator or designee will make every effort to regain contact with the subject after 1 year to collect the living situation 1 year after discharge (where possible, 3 telephone calls or by contacting the general practitioner).

**Clinician data.** All nurses, junior and senior doctors taking care of hospitalized patients in the 10 participating departments will be informed about the study on their nursing or medical staff meeting, those who give written electronic informed consent will be included in the study. Junior and senior doctors are coached for 4 months; nurses, junior and senior doctors give daily perceptions during the 12-months study period and fill-out a survey on personal characteristics and EDMCQ before and after the study period; head nurses and medical heads of departments fill out a questionnaire on department characteristics before and after the study period.

**Quality of the implementation of the intervention data.** To evaluate the quality of the implementation of coaching, process measures based on the RE-AIM implementation framework will be collected (see Table 4).

## Data management

An electronic case report form (eCRF) will be completed for each participant summarizing all study data. Subjects that are included in the study, will be assigned a unique study number upon their registration in REDCap electronic data capturing tool. Participants will only be recorded by their participant number. The subject identification list will be safeguarded by the site. The name and any other directly identifying details will not be included in the study database. The principal investigator and trial nurses will monitor department compliance via completion of the eCRF and feedback to the local investigators. Site visits conducted independently of the investigator team by regulatory agencies will be included (non-exhaustive list): review of informed consents and the followed process, check on recruitment status, checking for protocol deviations/violations, checking good clinical practice (GCP) compatibility, check on safety reporting compliance, IMP handling and storage and review of study data.

## Ethics

The protocol has been reviewed and approved by the Ethics Committee of the Ghent University Hospital (BC-09828, date of approval May 27th 2021). This committee includes patients' representatives. The study will be conducted according to the Declaration of Helsinki and the latest version of the International Council for Harmonisation (ICH) E6 (R2) GCP guidelines as adopted by the European Medicine Agency, creating a standard for the design, conduct, performance, monitoring, auditing, recording, analyses and reporting of clinical studies that provides assurance that the data and reported results are accurate and that the rights, integrity and confidentiality of study subjects are protected.

## Trial registration

ClinicalTrials NCT 05167019.

## Statistical analyses

Primary analysis will follow the intention-to-treat principle. Analysis of the incidence of written DNIR order will be based on logistic mixed effects models with random intercept to account for between-department variability, assuming a constant risk before and after intervention, and a linearly changing risk during the intervention.

While period effects are not expected, we will examine evidence for period effects in a secondary analysis by including a fixed linear time slope in the model; we will report the corresponding adjusted intervention effect. In a subsequent secondary analysis, we will use instrumental variable methods to account for noncompliance either due to clinicians not attending all coaching sessions, or due to clinicians previously having attended coaching sessions when they worked in a different department. In such analyses, we will in a first stage fit a linear model to predict the percentage of planned coaching sessions attended by clinicians as a function of time and randomized arm. In a second stage, a logistic mixed effects model will be fitted for the incidence of written DNIR order in function of time and the first-stage prediction. The coefficient of this first-stage prediction then expresses the effect of attending all coaching sessions under perfect adherence.

Analysis of the change in EDMCQ score will be based on linear mixed effects models for the EDMCQ score, an intervention indicator (1 if the clinician was already exposed to the intervention and 0 otherwise), a random intercept to account for between-department variability and a nested random intercept to account for between-clinician variability.

For secondary endpoints, the analysis of continuous endpoints will be based on linear mixed models and the analysis of dichotomous endpoints on logistic mixed models, each time including a random intercept at the ward level to account for between-department variability. Analyses at the patient level will include a fixed effect of time to correct for period effects, if there is evidence for such effects at the 5% significance level. Analyses at the clinician level will instead include a nested random intercept to account for between-clinician variability.

To acknowledge the multilevel structure, we will fit linear mixed models with two random intercepts: one for departments and one for clinicians.

## Discussion

This is the first stepped-wedge randomized control trial exploring the effects of coaching doctors in daily care, focusing on self-reflection and empowering leadership, and on the management of team dynamics, with regard to ethical decision-making in patients potentially receiving excessive treatment. In contrast to other investigators who focused on the effect of a consultant outside the team (communication facilitator, palliative or ethics or consultant. . .) [66–71], we preferred to perform an intervention inside the team because most often doctors still have to trigger or give their approval for consultations of experts from outside the team. We also decided to coach doctors in self-reflective and empowering leadership on the field via real repetitive patient situations because we think this will be more effective. This is in contrast with previous leadership-development programs that mainly targeted resident doctors or doctors in mid-level position and focused on skills training and technical and conceptual knowledge via workshops, lecture, plenary speeches or group discussion, respectively. Moreover, although all 45 studies report positive outcomes, few report system-level effects such as improved performance on quality indicators of customer satisfaction [72].

### Risk benefit of this intervention

The intervention is focused on doctors whereas the impact of the intervention will be measured at the patient, relatives and clinician levels. Given the consequences of providing excessive treatment at all levels (see background) our intervention has a favorable risk to benefit ratio.

1. The intervention is endorsed by the strategic quality cell and the medical council and is therefore completely embedded in the existing Ghent University Hospital structure. The safety, in case of for instance conflicts within the team, will be guaranteed by the HR department together with the coach supervisor. Potential conflicts with patients or relatives will be managed as usual, by the doctor in charge or the head of department in collaboration with the ombudsperson.

2. One could argue that this intervention may shorten patients' life. Firstly, although this may be an issue, it is important to note that the primary intention of this intervention is not to shorten life but to reduce excessive treatment, and as such, suffering of patients with a high risk of dying. By allowing clinicians to express anonymous perceptions of excessive treatment, timely identification of patients potentially receiving excessive treatment will certainly improve (sensitivity), however the quality of decision-making will be better guaranteed by enhancing reflection and decision-making in the team while taking the

wishes of patients and relatives into account (specificity). It is also of note that patient safety increases in such a setting because of direct communication and negotiation between doctor and patient, between doctors, and between doctors and nurses will expand. This will enhance fine-tuned decision-making for the benefit of the patient and will reduce avoidance behavior in complex clinical situations. Secondly, previous interventions aiming at reducing excessive treatment have shown to increase the quality of life in patients at risk of dying without shortening survival [66–71]. Because of all these reasons we decided to use written DNIR orders as primary endpoint and to consider mortality and survival with a good quality of live at home as secondary endpoints. Thirdly, written DNIR orders aim at clarifying to patients, relatives and the team what to do in case of deterioration and are therefore also not uniformly associated with mortality. Finally, we decided to power our intervention to detect an increase in written DNIR order from 35% to 50% and not higher, because this would not be a realistic expectation and because 7% of patients potentially receiving excessive treatment are still alive, at home with a good quality of life after 1 year [14].

## Supporting information

**S1 Appendix. SPIRIT 2013 checklist: Recommended items to address in a clinical trial protocol and related documents**\*.
(DOC)

**S2 Appendix. Coaching protocol.**
(PDF)

**S1 Protocol. IRB protocol and approval.**
(PDF)

## Acknowledgments

We wish to thank Anouska De Smeytere and Bram Gadeyne for respectively logistic and ICT support and all persons who contributed to the study as part of the CODE study group: Céline Jacobs and Aglaja De Pauw from the Medical Oncology Department, Alfred Meurs and Dimitri Hemelsoet from the Neurology Department, Jiska Maloteaux from the Internal Medicine Department, Anja Velghe and Nele Van Den Noorgate from the Geriatrics Department, Pieter Depuydt and Patrick Druwé from the Intensive Care Medicine Department, Hans Van Vlierberghe, Anja Geerts, Eduard Callebaut and Karen Geboes from the Gastroenterology and Hepatology Department, Eric Derom and Dieter Stevens from the Pneumology Department, Michel De Pauw, Lineke Hens and Fiona Tromp from the Cardiology Department, Ine Moors and Fritz Offner from the Hematology Department and Francis Verbeke and Wim Van Biesen from the Nephrology Department, all at the Ghent University Hospital, Belgium.

## Author Contributions

**Conceptualization:** Dominique D. Benoit, Stijn Vanheule, Frank Manesse, Frederik Anseel, Geert De Soete, Ruth Piers.

**Data curation:** Dominique D. Benoit.

**Formal analysis:** Dominique D. Benoit, Stijn Vansteelandt.

**Funding acquisition:** Dominique D. Benoit.

**Investigation:** Dominique D. Benoit.

**Methodology:** Dominique D. Benoit, Stijn Vanheule, Frank Manesse, Frederik Anseel, Geert De Soete, Katrijn Goethals, An Lievrouw, Stijn Vansteelandt, Erik De Haan, Ruth Piers.

**Project administration:** Dominique D. Benoit, Katrijn Goethals, Ruth Piers.

**Supervision:** Dominique D. Benoit, Stijn Vanheule, Frank Manesse, Erik De Haan, Ruth Piers.

**Writing – original draft:** Dominique D. Benoit.

**Writing – review & editing:** Dominique D. Benoit, Stijn Vanheule, Frank Manesse, Frederik Anseel, Geert De Soete, Katrijn Goethals, An Lievrouw, Stijn Vansteelandt, Erik De Haan, Ruth Piers.

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
