## [Decision Letter · Decision Letter 0]

7 Dec 2022

PONE-D-22-24774Coaching doctors to improve ethical decision-making in adult hospitalised patients potentially receiving excessive treatment: study protocol for a stepped wedge cluster randomised controlled trialPLOS ONE

Dear Dr. Benoit,

Thank you for submitting your manuscript to PLOS ONE. After careful consideration, we feel that it has merit but does not fully meet PLOS ONE’s publication criteria as it currently stands. Therefore, we invite you to submit a revised version of the manuscript that addresses the points raised during the review process.

We look forward to receiving your revised manuscript.

Kind regards,

Oathokwa Nkomazana, MD MSC PhD

Academic Editor

PLOS ONE

Journal Requirements:

Additional Editor Comments (if provided):

Thank you for submitting a very interesting protocol on a novel method to improve ethical decision making among doctors. Please respond to the reviewers' comments.

Reviewers' comments:

Reviewer's Responses to Questions

**Comments to the Author**

1. Does the manuscript provide a valid rationale for the proposed study, with clearly identified and justified research questions?

Reviewer #1: Yes

Reviewer #2: Partly

Reviewer #3: Yes

2. Is the protocol technically sound and planned in a manner that will lead to a meaningful outcome and allow testing the stated hypotheses?

Reviewer #1: Yes

Reviewer #2: Yes

Reviewer #3: Yes

3. Is the methodology feasible and described in sufficient detail to allow the work to be replicable?

Reviewer #1: Yes

Reviewer #2: Yes

Reviewer #3: Yes

4. Have the authors described where all data underlying the findings will be made available when the study is complete?

Reviewer #1: Yes

Reviewer #2: Yes

Reviewer #3: Yes

5. Is the manuscript presented in an intelligible fashion and written in standard English?

Reviewer #1: Yes

Reviewer #2: Yes

Reviewer #3: Yes

6. Review Comments to the Author

You may also provide optional suggestions and comments to authors that they might find helpful in planning their study.

Reviewer #1: Dear journal editor

Before embarking further on a review, I stumbled upon rule of acceptance nbr 1 which is copied here

1. The study presents the results of original research.

The study in the article has no results yet, the article is describing a trial in progress. Therefore it seems rule nbr1 is not fulfilled.

On the other hand , if you waive this rule nbr 1, I am happy to comment and accept with minor revision the study which is interesting in itself

Reviewer #2: This is a protocol paper describing a stepped-wedge design (SWRT) aimed to make system-level interventions to improve ethical decision-making by medical doctors. I have a few concerns that need to be addressed,

1. Picture in Figure 3 is extremely low quality and I can’t read it.

2. SWRT design described in Figure 3 seems not done properly as in SWRT once a unit goes to Intervention never comes back to its previous regime. But Figure 3 seems they revert back after 4 months’ time. That is not a typical SWRT.

3. The power analysis section needs further elucidation. For example, please mention which statistical model is used and what software is used to power the model. Also mention whether the primary analysis model and power analysis model is the same or different and why it is so. I.e. if the Linear mixed model is used for power analysis and using what program/software!!

3. Randomization in any Cluster randomized model is challenging and so does for SWRT. It is not clear who will perform it and where the randomization key will be kept.

4. Since SWRT is a longitudinal model, missing data is unavoidable. Any plan to handle that must be described. Also, describe its effect on power analysis.

5. Outcomes are described to come from various levels. This means this is an SWRT with a multi-level model. Statistical analysis model for such data is challenging, please describe some of it.

Reviewer #3: Thank you for the opportunity to review this manuscript describing a study protocol for a stepped-wedge cluster RCT (SWT) for a novel intervention which involves coaching doctors to improve ethical decision making for hospitalized adult patients identified as receiving potentially excessive treatment.

The authors adequately provide a description of the scientific and social value for the proposed study.

I have a few observations for the authors to consider;

1. In the abstract, there is need to specify the actual endpoint of the questionnaire as the questionnaire itself is not an endpoint but a measurement tool to evaluate an endpoint.

2. Details on how the pilot study that informed this proposed SWT was conducted are needed for the benefit of the reviewers and readers (Line 127-129)

3. The implementation domain of the RE-AIM framework is generally categorised into "feasibility", "acceptibility" or "fidelity". As an implementation outcome,"experiences and satisfaction" best aligns with "Acceptibility" and this needs to be stated as such. Given that the intervention is defined by a protocol, it would be important to also evaluate "fidelity" to the protocol as an implementation outcome. [Table 2c]

4. It is not clear what informed the assumption of intra-departmental correlation [Line 444, Line 456-458]. These assumptions need to be stated and cited where possible.

7. PLOS authors have the option to publish the peer review history of their article (what does this mean?). If published, this will include your full peer review and any attached files.

Reviewer #1: No

Reviewer #2: No

Reviewer #3: No

---

## [Author Response · Author response to Decision Letter 0]

5 Jan 2023

Dear editor,

We would like to thank the reviewers for sharing their expertise with us and for their useful comments. Please find below our answers to the comments. We have revised our manuscript accordingly.

We have also uploaded an excel file with the names and affiliations of the members of the CODE study group as such that their co-author / membership can be acknowledged in pubmed. 

Kind regards,

Dominique BENOIT on behalf of the CODE-study investigators. 

Reviewer 1 . The study presents the results of original research. The study in the article has no results yet, the article is describing a trial in progress. Therefore it seems rule nbr1 is not fulfilled. On the other hand , if you waive this rule nbr 1, I am happy to comment and accept with minor revision the study which is interesting in itself

Thank you for sharing your expertise with us. Indeed this article describes a trial in progress. We are not aware of the fact that we uploaded our protocol in the wrong manuscript category. We will double check this during the uploading of our revised manuscript 

Reviewer 2

This is a protocol paper describing a stepped-wedge design (SWRT) aimed to make system-level interventions to improve ethical decision-making by medical doctors. I have a few concerns that need to be addressed,

Thank you for sharing your expertise with us and for your comments. 

1. Picture in Figure 3 is extremely low quality and I can’t read it. 

This is now a reconversion from an excel figure to tiff format which was checked in PACE. We changed the colours a bit to improve the readability. We hope the readability is better now

2. SWRT design described in Figure 3 seems not done properly as in SWRT once a unit goes to Intervention never comes back to its previous regime. But Figure 3 seems they revert back after 4 months’ time. That is not a typical SWRT.

We believe that the label of SWRT is also appropriate here since we evaluate the effect of initiating the coaching intervention from a given point in time onwards. In our analysis, we do not `pretend’ as if the intervention comes back to the original regime, as there may – hopefully – be a lasting effect of the intervention. This analysis aligns with the usual intention-to-treat analysis.

3. The power analysis section needs further elucidation. For example, please mention which statistical model is used and what software is used to power the model. Also mention whether the primary analysis model and power analysis model is the same or different and why it is so. I.e. if the Linear mixed model is used for power analysis and using what program/software!!

The statistical model is stated on lines 430-441. The phrase `Based on such analysis’ on line 432 confirms that the same model is used for the analysis and the power calculation. The software used is RStudio 2022.07.2+576 "Spotted Wakerobin" for macOS Mozilla/5.0. The paper states that a Monte Carlo (i.e., simulation-based) power calculation was made. No specific software is available for this is; instead, the power calculation was programmed in R. On Page 24 Lines 467-468:

We added ‘The software used is RStudio 2022.07.2+576 "Spotted Wakerobin" for macOS Mozilla/5.0.

4. Randomization in any Cluster randomized model is challenging and so does for SWRT. It is not clear who will perform it and where the randomization key will be kept.

We agree that this is challenging, that is why expert statistical advise was included for the design of the study. This is also stated in the manuscript on Page 25 Lines 471-473:

Randomisation of the 10 departments was performed by the Ghent University Department of Applied Mathematics, Computer Science and Statistics based on a random number generator in the software R.

5. Since SWRT is a longitudinal model, missing data is unavoidable. Any plan to handle that must be described. Also, describe its effect on power analysis.

We did not describe the effect of missing data on the power analysis because the expected sample size is 1680 patients, but only 605 patients are required. The planned analysis based on a linear mixed model adjusts for missing outcome data, provided that the missingness is `missing at random’.

6. Outcomes are described to come from various levels. This means this is an SWRT with a multi-level model. Statistical analysis model for such data is challenging, please describe some of it.

To acknowledge the multilevel structure, we will fit linear mixed models with 2 random intercepts: 1 for department and 1 for clinician. 

Because such models are fairly standard, we are unsure what additional detail the reviewer is requesting. We added on page 28, Lines 544-545:

‘To acknowledge the multilevel structure, we will fit linear mixed models with two random intercepts: one for departments and one for clinicians.’

Reviewer 3

Thank you for the opportunity to review this manuscript describing a study protocol for a stepped-wedge cluster RCT (SWT) for a novel intervention which involves coaching doctors to improve ethical decision making for hospitalized adult patients identified as receiving potentially excessive treatment.

The authors adequately provide a description of the scientific and social value for the proposed study.

I have a few observations for the authors to consider;

Thank you for sharing your expertise with us and for your comments.

1. In the abstract, there is need to specify the actual endpoint of the questionnaire as the questionnaire itself is not an endpoint but a measurement tool to evaluate an endpoint.

We changed ‘ethical decision-making climate questionnaire’ to ‘quality of ethical decision-making climate’in the abstract

2. Details on how the pilot study that informed this proposed SWT was conducted are needed for the benefit of the reviewers and readers (Line 127-129)

Thank you. We added some more explanation on the way the pilot study was conducted.

In order to perform the power analysis for the current study, we measured the incidence of patients with two or more perceptions of excessive care by different clinicians in the wards of the Ghent University Hospital willing to participate in this study. 

A dedicated researcher actively surveyed all nurses and doctors during one week on every department asking for which patients the clinician was responsible and in which patients they perceived the care as excessive. The DNIR code for these patients was retrieved from the head nurse.

Of the 258 patients admitted in these wards, 32 (12%) were perceived as receiving excessive care by two or more different clinicians and only 12 (38%) had a written DNIR order (Table 1). Page 7, Lines 129-132: We added ‘A dedicated researcher actively surveyed all nurses and doctors during one week on every department asking for which patients the clinician was responsible and in which patients they perceived the care as excessive. The DNIR code for these patients was retrieved from the head nurse.’

3. The implementation domain of the RE-AIM framework is generally categorised into "feasibility", "acceptibility" or "fidelity". As an implementation outcome,"experiences and satisfaction" best aligns with "Acceptibility" and this needs to be stated as such. Given that the intervention is defined by a protocol, it would be important to also evaluate "fidelity" to the protocol as an implementation outcome. [Table 2c]

Thank you for the remark. We adapted Table 2C on page 17 accordingly: 

(1) We added as outcome for adoption, implementation and maintenance: ‘experiences and satisfaction with the intervention; fidelity to the protocol and long-term perspectives’

(2) We changed the Legend accordingly to:

*Process measures based on RE-AIM framework, ** Adoption = experiences with the intervention, Implementation = feasibility, acceptability (experiences and satisfaction) and fidelity to the protocol, Maintenance = long-term adaptation of the intervention

4. It is not clear what informed the assumption of intra-departmental correlation [Line 444, Line 456-458]. These assumptions need to be stated and cited where possible.

Based on the data from the pilot study, we first fitted a logistic mixed effects model for the incidence of disproportionate care. This resulted in a between-unit standard deviation of 0.2385, which we rounded to 0.25 for the power calculation. This corresponds with an intraclass correlation of 0.25/(0.25+3.14152)=0.0247, which we rounded to 0.025. Page 23, Lines 446-449: We changed ‘In the power analysis, we assumed an intra-department correlation of 0.025.’to ‘Based on the data from the pilot study, we first fitted a logistic mixed effects model for the incidence of excessive treatment. This resulted in a between-unit standard deviation of 0.25, corresponding to an intraclass correlation of 0.025, used for the power analyses.’

---

## [Editor Report · Decision Letter 1]

24 Jan 2023

Coaching doctors to improve ethical decision-making in adult hospitalised patients potentially receiving excessive treatment: study protocol for a stepped wedge cluster randomised controlled trial

PONE-D-22-24774R1

Dear Dr. Benoit,

We’re pleased to inform you that your manuscript has been judged scientifically suitable for publication and will be formally accepted for publication once it meets all outstanding technical requirements.

Kind regards,

Oathokwa Nkomazana, MD MSC PhD

Academic Editor

PLOS ONE
---

## [Editor Report · Acceptance letter]

31 Jan 2023

PONE-D-22-24774R1 

Coaching doctors to improve ethical decision-making in adult hospitalised patients potentially receiving excessive treatment: study protocol for a stepped wedge cluster randomised controlled trial 

Dear Dr. Benoit:

I'm pleased to inform you that your manuscript has been deemed suitable for publication in PLOS ONE. Congratulations! Your manuscript is now with our production department. 

Kind regards, 

on behalf of

Dr. Oathokwa Nkomazana 

Academic Editor

PLOS ONE